# Implementation of a New Strongly-Asymmetric Algorithm and Its Optimization

**Koki Jimbo [1],\* , Satoshi Iriyama [1] and Massimo Regoli [2]** 

1  Information Science Department, Tokyo University of Science, 2641 Yamazaki, Noda, Chiba 278-0022, Japan; iriyama@is.noda.tus.ac.jp

2  DICII, Engineering Faculty Via del Politecnico Universitá di Roma Tor Vergata, 1, 00133 Roma, Italy; regoli@uniroma2.it

\*  Correspondence: 6319702@ed.tus.ac.jp

**Abstract:** A new public key agreement (PKA) algorithm, called the strongly-asymmetric algorithm (SAA-5), was introduced by Accardi et al. The main differences from the usual PKA algorithms are that Bob has some independent public keys and Alice produces her public key by using some part of the public keys from Bob. Then, the preparation and calculation processes are essentially asymmetric. This algorithms has several free parameters more than the usual symmetric PKA algorithms and the velocity of calculation is largely dependent on the parameters chosen; however, the performance of it has not yet been tested. The purpose of our study was to discuss efficient parameters to share the key with high speeds in SAA-5 and to optimize SAA-5 in terms of calculation speed. To find efficient parameters of SAA-5, we compared the calculation speed with Diffie–Hellman (D-H) while varying values of some parameters under the circumstance where the length of the secret shared key (SSK) was fixed. For optimization, we discuss a more general framework of SAA-5 to find more efficient operations. By fixing the parameters of the framework properly, a new PKA algorithm with the same security level as SAA-5 was produced. The result shows that the calculation speed of the proposed PKA algorithm is faster than D-H, especially for large key lengths. The calculation speed of the proposed PKA algorithm increases linearly as the SSK length increases, whereas D-H increases exponentially.

**Keywords:** public key exchange; security; asymmetric; asymmetric algorithm; cryptography; framework; generalization

## 1. Introduction

The discovery of the Diffie–Hellman(D-H) [1] public key agreement (PKA) protocol and RSA [2] asymmetric cryptography are two of the greatest achievements for the literature of data protection. Although it has been over 40 years since the discovery of those algorithms, they are still utilized not only for key agreement but also for various scenes (e.g., digital signature) along with algorithms such as ElGamal [3], Elliptic curve D-H [4], etc.

However, a lot of threats were developed at the same time. Because of recent increase in the computational power of eavesdroppers, the small key lengths of D-H or RSA are no longer safe [5]. Even for longer keys, these algorithms are expected to become vulnerable in the near future because of Shor's quantum algorithm for both the integer factorization problem and discrete logarithm problem [6].

As a solution against these threats, studies of a modern PKA and asymmetric cryptography are widely spread. Algorithms based on multivariate polynomial equations [7,8] and lattices [9,10] are the most well known ones. These algorithms utilize matrices and security, which are based

on NP-hard problems, e.g., the difficulty of solving a system of multivariate quadratic polynomial equations and the difficulty of the shortest vector problem. In 2019, NIST announced 26 public key cryptographic algorithms as candidates for the standardization of post-quantum-cryptographic systems [11]. The lattice based ones, such as NewHope [12] and NTRU [13], and the multivariate polynomial based digital signature algorithms such as GeMSS [14] are included in the list.

In [15], Accardi et al. proposed a new scheme of public key agreement based on non-commutative algebra called a strongly asymmetric public key agreement (SAPKA). Because this scheme is very general, we need concrete realizations to estimate computational and breaking complexity. In [16], strongly-asymmetric algorithm 3 (SAA-3) is introduced as one of the concrete realizations of SAPKA. SAA-3 is based on matrix algebra with element-wise matrix exponentiation, which is called the Schur-product. Strongly-asymmetric algorithm 5 (SAA-5), introduced in [17], has a similar structure to SAA-3, but there are differences. The main differences between them are:

- The public parameter $\alpha$ is removed.
- The secret key of Alice is scalar in [16]. In [17], this is replaced by a matrix set.
- The constraints on Bob's secret keys are reduced to the requirement that certain matrices should not be invertible.

Strictly speaking, SAA-5 cannot be described in the form of SAPKA because of the secret key structure of Alice, but both computational and breaking complexity are improved from SAA-3.

In this paper, we explain the mathematical setting of SAA-5 and the breaking strategy introduced in [17] and report our performance test compared with D-H. After that, we tried to realize a much faster PKA than SAA-5 by considering the general PKA class that SAA-5 belongs to and has much more freedom than SAPKA in term of the key structure of Alice.

## 2. Mathematical Setting and Key Agreement Protocol of SAA-5

The key agreement process between Alice and Bob are:

**Step 1.** Alice and Bob share following public information:

$$\text{a natural integer } d \in \mathbb{N} \,,$$

$$\text{a finite field } \mathbb{F} := \mathbb{Z}_p \text{ where p is a large prime number },$$

$$\text{a finite set } I \subset \mathbb{N} \,.$$

**Step 2.** Bob creates his secret keys as matrices:

$$x_B \in M(d; \mathbb{F}) \,,$$

$$N_B \in M(d; \mathbb{F}) \,,$$

$$c \in \mathbb{F} \,,$$

and as a set of matrices:
$$\mathbb{A} := \{ A_j \in M(d, \mathbb{F}) \,, j \in I \} \,.$$

For all secret keys of Bob, the following conditions must be satisfied:
- $N_B$ must be invertible
- Each $A_j$ ($j \in I$) must not be invertible.

**Step 3.** Bob creates his public keys for all $j \in I$ as:

$$y_{B,2;j} := c^{\circ(A_j N_B)} \in M(d; \mathbb{F}) \,,$$

$$y_{B,3;j} := c^{\circ(A_j x_B)} \in M(d; \mathbb{F}) \,.$$

Here, the symbol $c^{\circ M}$ denotes the matrix:

$$c^{\circ M} := \left( c^{M_{a,g}} \right) \qquad ; \qquad a, g \in \{1, \cdots, d\}, \tag{1}$$

which is called *the Schur exponentiation of c by M*.

**Step 4.** Alice creates her secret key as a matrix set as:

$$x_A := \{x_{A,j} \in M(d, \mathbb{F}), j \in I\},$$

and creates her public key $y_A \in M(d, \mathbb{F})$ using one of Bob's public key $y_{B,2;j}$ as: For each $a, g \in \{1, \cdots, d\}$,

$$
(y_{A;a,g}) = \left( \prod_{j \in I} \prod_{b \in \{1,\cdots,d\}} (y_{B,2;j})_{b,g}^{(x_{A,j})_{a,b}} \right) = \left( \prod_{j \in I} \prod_{b \in \{1,\cdots,d\}} (c^{A_j N_B})_{b,g}^{(x_{A,j})_{a,b}} \right)
$$

$$
= \left( \prod_{j \in I} \prod_{b \in \{1,\cdots,d\}} (c^{(x_{A,j})_{a,b}(A_j N_B)_{b,g}}) \right) = \left( c^{\sum_{j \in I} \sum_{b \in \{1,\cdots,d\}} (x_{A,j})_{a,b}(A_j N_B)_{b,g}} \right)
$$

$$
= \left( c^{(\sum_{j \in I} x_{A,j} A_j N_B)_{a,g}} \right) = c^{\circ \sum_{j \in I} x_{A,j} A_j N_B}.
$$

**Step 5.** Bob computes his secret shared key (SSK) $\kappa_B \in M(d, \mathbb{F})$ using the public key of Alice and his own secret keys $x_B$ and $N_B$ as: For each $a, g \in \{1, \cdots, d\}$,

$$
\kappa_B = \left( \prod_{b \in \{1,\cdots,d\}} (y_A)_{a,b}^{(N_B^{-1} x_B)_{b,g}} \right) = \left( \prod_{b \in \{1,\cdots,d\}} (c^{\circ \sum_{j \in I} x_{A,j} A_j N_B})_{a,b}^{(N_B^{-1} x_B)_{b,g}} \right).
$$

**Step 6.** Alice computes her SSK $\kappa_A \in M(d, \mathbb{F})$ by using $y_{B,3;j}$ and her own secret key $x_A$ as:

$$
\kappa_A = \left( \prod_{j \in I} \prod_{b \in \{1,\cdots,d\}} (y_{B,3;j})_{b,g}^{(x_{A,j})_{a,b}} \right) = \left( \prod_{j \in I} \prod_{b \in \{1,\cdots,d\}} (c^{A_j x_B})_{b,g}^{(x_{A,j})_{a,b}} \right).
$$

The equality of $\kappa_A$ and $\kappa_B$ is guaranteed by the following equations. The SSK of Alice is:

$$
\kappa_A = \left( \prod_{j \in I} \prod_{b \in \{1,\cdots,d\}} (c^{A_j x_B})_{b,g}^{(x_{A,j})_{a,b}} \right) = \left( \prod_{j \in I} \prod_{b \in \{1,\cdots,d\}} (c^{(x_{A,j})_{a,b}(A_j x_B)_{b,g}}) \right)
$$

$$
= \left( c^{\sum_{j \in I} \sum_{b \in \{1,\cdots,d\}} (x_{A,j})_{a,b}(A_j x_B)_{b,g}} \right) = \left( c^{\sum_{j \in I} x_{A,j} A_j x_{B_{a,g}}} \right)
$$

$$
= c^{\circ \sum_{j \in I} x_{A,j} A_j x_B}. \tag{2}
$$

and the SSK of Bob is:

$$
\kappa_B = \left( \prod_{b \in \{1,\cdots,d\}} (c^{\sum_{j \in I} x_{A,j} A_j N_B})_{a,b}^{(N_B^{-1} x_B)_{b,g}} \right)
$$

$$
= \left( \prod_{b \in \{1,\cdots,d\}} c^{(\sum_{j \in I} x_{A,j} A_j N_B)_{a,b}(N_B^{-1} x_B)_{b,g}} \right) = \left( c^{\sum_{b \in \{1,\cdots,d\}} (\sum_{j \in I} x_{A,j} A_j N_B)_{a,b}(N_B^{-1} x_B)_{b,g}} \right)
$$

$$
= \left( c^{(\sum_{j \in I} x_{A,j} A_j N_B N_B^{-1} x_B)_{a,g}} \right) = \left( c^{(\sum_{j \in I} x_{A,j} A_j x_B)_{a,g}} \right)
$$

$$= c^{\circ \sum_{j \in I} x_{A,j} A_j x_B} . \tag{3}$$

Obviously (2) = (3).

## 2.1. Breaking Complexity of SAA-5

The breaking complexity of SAA-5 is already discussed in [17]. The eavesdropper(Eve) tries to recover the SSK by using the following public parameters:

- Common parameters $d$, $p$, $I$.
- Public keys of Bob $y_{B,2;j}$, $y_{B,3;j}$ for all $j \in I$.
- Public key of Alice $y_A$.

Alice knows the following values for all $j \in I$, denoting $x_1 := \sum_{j \in I} x_{A,j} A_j$, $x_2 := N_B$, $x_{3,j} := A_j$, $x_4 := x_B$ and $\log c$:

$$\alpha_1 = \log y_A = \sum_{j \in I} x_{A,j} A_j N_B \log c = x_1 x_2 \log c , \tag{4}$$

$$\alpha_{2,j} = \log y_{B,2;j} = A_j N_B \log c = x_{3,j} x_2 \log c , \tag{5}$$

$$\alpha_{3,j} = \log y_{B,3;j} = A_j x_B \log c = x_{3,j} x_4 \log c . \tag{6}$$

But, for Eve to get $\alpha_1$, $\alpha_{2,j}$ and $\alpha_{3,j}$, she needs to solve the discrete logarithm problem for each value. Here, we assume the cost for solving discrete logarithm problem is 0 so, Eve can immediately calculate all $\alpha_1$, $\alpha_{2,j}$ and $\alpha_{3,j}$ in this case.

After getting all $\alpha_1$, $\alpha_{2,j}$ and $\alpha_{3,j}$, her strategy is to calculate $\log \kappa$ from the equation:

$$\log \kappa = \sum_{j \in I} x_{A,j} A_j x_B \log c = x_1 x_4 \log c , \tag{7}$$

by deducing $x_1$, $x_2$, $x_{3,j}$, $x_4$ and $\log c$ from (4)–(6). This system contains $4d^2 + 1$ unknowns for $3d^2$ number of polynomials so, $x_1, x_2, x_{3,j}, x_4$ and $\log c$ are not determined uniquely. we try to brute-force attack one unknown of the above system to estimate the breaking complexity of the algorithm. As an example, we try this on $x_2$ and $S_{bf;x_2}$ denotes the space she has to search for this attack.

## 2.2. The Brute-Force Attack

Eve has as the same number of choices as the cardinality of $GL(d; \mathbb{F})$ for $x_2$. If she finds $x_2$ equal to $N_B$, then $x_1 \log c = \sum_{j \in I} x_{A,j} A_j \log c$ and $x_{3,j} \log c = A_j \log c$ are satisfied from (4) and (5) because $x_2 = N_B$ is invertible. From (6), she wants to know $x_4$ to satisfy $x_4 = x_B$, however, $x_{3,j} \log c$ is not an invertible matrix and the solution of (6) contains $d^2 - rank(x_{3,j} \log c)$ number of arbitrary elements. This means one from $p^{d^2 - rank(x_{3,j} \log c)}$ number of candidates satisfies $x_4 = x_B$ so, $S_{bf;x_2}$ is:

$$S_{bf;x_2} = |GL(d; \mathbb{F})| p^{d^2 - rank(x_{3,j} \log c)} , \tag{8}$$

which is extremely large even if $p$ is relatively small such as 16 bit or 32 bit. Moreover, she cannot judge whether $x_2 = N_B$ and $x_4 = x_B$ are satisfied because $N_B$ and $x_4$ are kept secret by Bob. This fact shows that even an exhaustive search is impracticable for this strategy.

For other security analyses, please refer to the attacks in Section 4 of [17], which more adequately shows the difficulty of the attacks.

## 3. Performance Estimation and Evaluation of SAA-5

We already know the difficulty in breaking the above-mentioned algorithm, however, the high-speed calculation for generating the SSK is also needed from the view point of practicality. Here, we estimate the time spent to generate the SSK and report our performance tests of SAA-5 verses D-H.

### 3.1. Performance Estimation

The computational complexity of SAA-5 is already analyzed in [17]. We show the estimated total multiplication steps needed to share the SSK.

The exponentiation of elements in $\mathbb{Z}_p$ needs $\log_2 p$ steps of multiplication in the worst case. Then, the Schur-exponential:

$$(c^{\circ M})_{a,g} = (c^{M_{a,g}})_{a,g} \text{,}$$

requires $d^2 \log_2 p$ steps of multiplication. The calculation:

$$\left( \prod_{b \in \{1, \cdots, d\}} (A_{a,b})^{B_{b,g}} \right) \text{,}$$

requires $d^2(d + \log_2 p) = d^3 + d \log_2 p$ steps of multiplication. Therefore, the calculation:

$$\left( \prod_{j \in I} \prod_{b \in \{1, \cdots, d\}} (A_{a,b})^{B_{b,g}} \right) \text{,}$$

requires $(|I| - 1)d^2(d + \log_2 p) = (|I| - 1)(d^3 + d^2 \log_2 p)$ steps of multiplication. Hence, we can estimate the multiplication steps needed for generating the SSK as the Table 1.

**Table 1.** Key size and estimation of the time for multiplication of keys.

| Key | Bit Size | Steps |
|-----|----------|-------|
| $y_A$ | $d^2 \log p$ | $(|I| - 1)(d^3 + d^2 \log p)$ |
| $\kappa_A$ | $d^2 \log p$ | $(|I| - 1)(d^3 + d^2 \log p)$ |
| $y_{B2}$ | $d^2 I \log p$ | $|I|(d^3 + d^2 \log p)$ |
| $y_{B3}$ | $d^2 I \log p$ | $|I|(d^3 + d^2 \log p)$ |
| $\kappa_B$ | $d^2 \log p$ | $2(d^3 + d^2 \log p)$ |
| Total | | $4I(d^3 + d^2 \log p)$ |

Note that the bit length of element in $\mathbb{Z}_p$ is expressed as $\lceil \log_2 p \rceil$, so, the bit length of modular matrix in $M(d, \mathbb{Z}_p)$ is expressed as $d^2 \lceil \log_2 p \rceil$ bits. As can be seen, the calculation steps is expected to be on the order of $d^3$.

### 3.2. Discussion: Efficient Parameters of SAA-5 and Comparison with D-H

Here, we report our performance test of SAA-5 versus D-H. Hereafter, We implement all PKA algorithms in the following environment:

- macOS Mojave ver10.14.6.
- 1.3 GHz Intel Core i5.
- 8 GB 1867 MHz LPDDR3.
- Language: JAVA.

We compare SAA-5 with D-H by fixing the length of the SSK. When the key length of the SSK is fixed, denoted by $\bar{\kappa}$, the total multiplication steps for sharing the SSK in SAA-5 (denoted by $CC_{SAA-5}$) is described as:

$$CC_{SAA-5} = 4|I|(d^3 + \bar{\kappa}) \text{,}$$

because $\bar{\kappa} = d^2 \log p$. In this case, the total computational complexity of D-H (denoted by $CC_{D-H}$) is described as:

$$CC_{D-H} \simeq 4\bar{\kappa} \,,$$

because Alice and Bob calculate a public key and their own SSK for each and the bit size of each secret key is the same size as that of the SSK, which is $\bar{\kappa}$. Since $\bar{\kappa}$ is constant, the calculation time of SAA-5 is expected to depend only on the parameter $d$. However, the fact that the time needed for calculating scalar exponentiation in $\mathbb{Z}_p$ depends on the size of $p$ and increases at an exponential rate is well known (for example, see pp. 5–6 of [18]). Therefore, we expect that the calculation time depends not only on $d$ but also on $p$. Before comparing SAA-5 with D-H, we show our experimental result, which shows how the parameters $p$ and $d$ effect the calculation speed of SAA-5, while the SSK length is fixed (16,384 bits).

The white circle in Figure 1 indicates the total time spent calculating the SSK for Alice and Bob, and the black diamond indicates the bit length of the SSK. The pair of dimension and bit length of Figure 1 are:

$$(d, \log p) = (2, 4096), (4, 1024), (8, 256), (16, 64), (32, 16) \,.$$

Roughly speaking, the time needed to share the SSK can be reduced by decreasing $p$ and increasing $d$ while keeping the length of SSK.

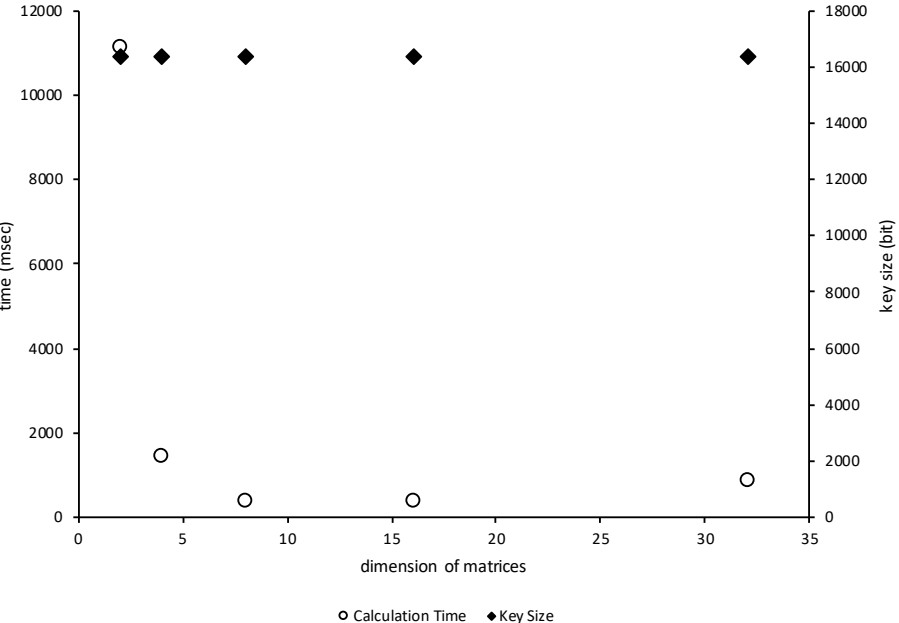

**Figure 1.** Changing time to compute a fixed-length key when $d$ is a variable.

Finally, we show the experimental result in which we compare SAA-5 with D-H. Table 2 and Figure 2 shows our result of comparison with D-H. Java codes of the algorithms can be referred in [19].

**Table 2.** Comparison of the time to generate the secret shared key (SSK).

| SSK Length (bit) | SAA-5 (msec) | D-H (msec) |
|---|---|---|
| 512 | 12.45 | 1.45 |
| 1024 | 20.63 | 3.37 |
| 1536 | 16.19 | 10.87 |
| 2048 | 18.10 | 24.21 |
| 2560 | 28.77 | 45.68 |
| 3072 | 23.54 | 83.39 |
| 3584 | 23.35 | 120.29 |
| 4096 | 24.42 | 219.90 |
| 4608 | 38.12 | 332.46 |
| 5120 | 39.58 | 620.86 |

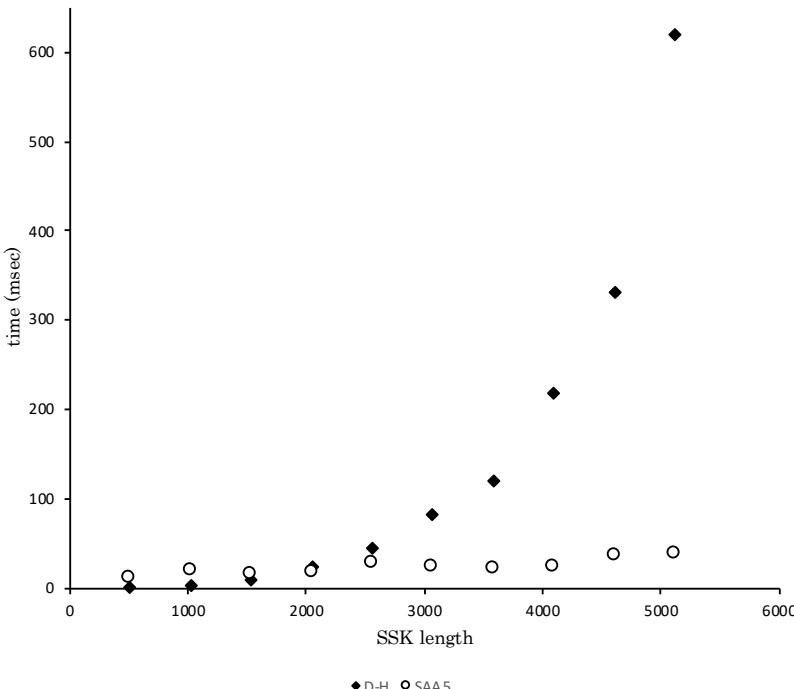

**Figure 2.** Comparison of the time to generate SSK.

*3.3. Experimental Result: Comparison with D-H*

In this experiment, we fixed the SAA-5 parameters as $d = 8$ and $|I| = 5$ so, the size of the prime number (denoted by $p_{SAA-5}$) is given by $\bar{\kappa}/64$ because the length of the SSK $\bar{\kappa}$ is expressed as $d^2 \log p$. The size of each public key of D-H is the same size as that of the SSK, which is $\bar{\kappa}$, so as a bit length of the prime. The bit size of the prime used in SAA-5 and D-H (denoted by $p_{D-H}$) is shown in Table 3. As can be seen from Table 2 and Figure 2, when the length of the SSK is over 2048 bit, the SAA-5 out-performs D-H, and the larger the SSK becomes, the the larger the difference of calculation speed between them becomes (SAA-5 is approximately 15 times faster than D-H when the SSK is 5120 bit).

**Table 3.** Comparison of the bit size of the prime number.

| SSK Length (bit) | $p_{SAA-5}$ (bit) | $p_{SAA-5}$ (bit) |
| --- | --- | --- |
| 512 | 8 | 512 |
| 1024 | 16 | 1024 |
| 1536 | 24 | 1536 |
| 2048 | 32 | 2048 |
| 2560 | 40 | 2560 |
| 3072 | 48 | 3072 |
| 3584 | 56 | 3584 |
| 4096 | 64 | 4096 |
| 4608 | 72 | 4608 |
| 5120 | 80 | 5120 |

## 4. Performance Improvement

We showed that the performance of SAA-5 is more effective than the D-H, especially for large keys because of its algebraic property and variety of parameters.

In this section, we try to realize a faster PKA with the same breaking complexity as SAA-5 by the following steps. As a first step, we extract the condition from SAA-5 that allows Alice and Bob to create their SSK, that is, we construct a new PKA framework similar to SAPKA [15] in which key agreement process is described by compositions of maps. As a second step, we fix the parameters (maps) of the framework to produce a new PKA algorithm. Then, we check if the new PKA algorithm possesses the same security level as and more efficient than SAA-5.

### 4.1. The SAPKA with Multiple Keys Class

Here, we define the new class called SAPKA with Multiple Keys class (SAPKA-MK) . The main property of this class is that Alice produces her secret key as a set $\{x_{A,j} \in \mathcal{P} \; ; \; j \in I\}$, not a single element as [15].

The SAPKA-MK algorithms have the following common ingredients:

- a semigroup $\mathcal{P}$ together with some operation $\bullet$.
- a set $\widehat{M}_{\mathcal{P}}$ of easily invertible maps $\mathcal{P} \to \mathcal{P}$, called *noise space*.
- a set $M_{\mathcal{P}}$ of maps : $\mathcal{P} \to \mathcal{P}$.
- a finite set $I := \{i_1, i_2, \ldots, i_n\} \subset \mathbb{N}$.

where $|I| = n$ .

Of these ingredients $\mathcal{P}$ is public and $M_{\mathcal{P}}$, $\widehat{M}_{\mathcal{P}}$ belong to the secret keys of $B$. The set

$$\mathcal{K}_B := \{M_{\mathcal{P}}\} \times M_{\mathcal{P}} \times M_{\mathcal{P}} \times \{M_{\mathcal{P}}\} \times \widehat{M_{\mathcal{P}}} \times \widehat{M_{\mathcal{P}}} \, ,$$

is used by Bob to construct his secret and public keys according to the following scheme. All maps are defined by a finite set of parameters, so to *send a map* means to send the corresponding set of parameters and the rules to combine them. Unless explicitly mentioned, an equality between two functions means that these functions have the same domain of definition and equality holds on it.

**Definition 1.** *Let $\mathcal{S}$ be a semigroup together with some operation $\bullet$ and let the functions be*

$$y_{1,j}, y_2, y_3, y_{4,j} \; : \; \mathcal{S} \to \mathcal{S} \, ,$$

for all $j \in I := \{i_1, i_2, \ldots, i_n\} \subset \mathbb{N}$. *The ordered quadruple* $(\{y_{1,j} ; j \in I\}, y_2, y_3, \{y_{4,j} ; j \in I\})$ *is said to satisfy the* **multiple compatibility condition** *if the following equation:*

$$(y_{1,i_1} \circ y_2(x_{i_1})) \bullet (y_{1,i_2} \circ y_2(x_{i_2})) \bullet \cdots \bullet (y_{1,i_n} \circ y_2(x_{i_n}))$$

$$= y_3(y_{4,i_1}(x_{i_1}) \bullet y_{4,i_2}(x_{i_2}) \bullet \cdots \bullet y_{4,i_n}(x_{i_n})) \, , \tag{9}$$

*is satisfied for all* $x_j$ ($j \in I$). *Hereafter, above symbol* $\circ$ *denotes map composition unless otherwise specified. If the condition* (9) *is satisfied only on a sub-semigroup* $\mathcal{S}_0 \subset \mathcal{S}$, *one says that the multiple compatibility condition* **is satisfied on** $\mathcal{S}_0$.

The key agreement for an algorithm belongs to SAPKA-MK is performed as:

**Step 1.** Bob chooses maps $(\{y_{1,j} ; j \in I\}, y_2, y_3, \{y_{4,j} ; j \in I\}, N_1, N_2) \in \mathcal{K}_B$. By conjugating $(y_{1,j}, y_2, y_3, y_{4,j}, N_1, N_2)$ for each $j \in I$, Bob constructs quadruple:

$$(\{y'_{1,j} ; j \in I\}, y'_2, y'_3, \{y'_{4,j} ; j \in I\}) \, ,$$

satisfies (9). and he uses $y'_3$ as secret key of him as a function:

$$x_B := y'_3 \, ,$$

and public keys $(y_{B,1}, y_{B,2})$ as following functions:

$$y_{B,1}(x_{i_1}, x_{i_2}, \ldots, x_{i_n}) := (y'_{1,i_1} \circ y'_2(x_{i_1})) \bullet (y'_{1,i_2} \circ y'_2(x_{i_2})) \bullet \cdots \bullet (y'_{1,i_n} \circ y'_2(x_{i_n})) \, , \tag{10}$$

$$y_{B,2}(x_{i_1}, x_{i_2}, \ldots, x_{i_n}) := y'_{4,i_1}(x_{i_1}) \bullet y'_{4,i_2}(x_{i_2}) \bullet \cdots \bullet y'_{4,i_n}(x_{i_n}) \, . \tag{11}$$

**Step 2.** Bob sends the public keys $(y_{B,1}, y_{B,2})$ to Alice.

**Step 3.** Alice chooses her secret key as a set $\{x_{A,j} \in \mathcal{P} ; j \in I\}$ then, she constructs her public key as an element:

$$y_A := y_{B,2}(x_{A,i_1}, x_{A,i_2}, \ldots, x_{A,i_n}) \in \mathcal{P} \, .$$

**Step 4.** Alice sends the public key $y_A$ to Bob.

**Step 5.** Alice computes the secret shared key (SSK) $\kappa$ as:

$$\kappa = y_{B,1}(x_{A,i_1}, x_{A,i_2}, \ldots, x_{A,i_n}) = (y'_{1,i_1} \circ y'_2(x_{A,i_1})) \bullet (y'_{1,i_2} \circ y'_2(x_{A,i_2})) \bullet \cdots \bullet (y'_{1,i_n} \circ y'_2(x_{A,i_n})) \, .$$

**Step6.** Bob computes $\kappa$ as:

$$\kappa = x_B(y_A) = y'_3(y'_{4,i_1}(x_{A,i_1}) \bullet y'_{4,i_2}(x_{A,i_2}) \bullet \cdots \bullet y'_{4,i_n}(x_{A,i_n})) \, .$$

**Remark 1.** *When* $n = 1$, *SAPKA-MK is equivalent to the SAPKA algorithm. This means that SAPKA-MK is one of general forms of SAPKA in terms of the number of Alice's secret key.*

Here, we introduce a lemma, which provides a constructive way to produce families to satisfy (9).

**Lemma 1.** *Let* $\mathcal{S}$ *be as Definition* 1 *and functions given as:*

$$y_{1,j}, y_2, y_3, y_{4,j} : \mathcal{S} \to \mathcal{S} \, ,$$

*for all* $j \in I$. *If the conditions:*

1.  *Each* $(y_{1,j}, y_2, y_3, y_{4,j})$ *satisfies the equation:*

$$y_{1,j} \circ y_2 = y_3 \circ y_{4,j} \tag{12}$$

2. $y_3 \in \{$*semi-group endomorphism* $\mathcal{S} \to \mathcal{S}\}$

*are satisfied for all $j \in I$, (9) is achieved.*

**Proof.** From the first condition, the equation:

$$(y_{1,i_1} \circ y_2(x_{i_1})) \bullet (y_{1,i_2} \circ y_2(x_{i_2})) \bullet \cdots \bullet (y_{1,i_n} \circ y_2(x_{i_n}))$$

$$= (y_3 \circ y_{4,i_1}(x_{i_1})) \bullet (y_3 \circ y_{4,i_2}(x_{i_2})) \bullet \cdots \bullet (y_3 \circ y_{4,i_n}(x_{i_n})),$$

is obvious for all $x_j$ ($j \in I$). By the endomorphism property of $y_3$ on $\mathcal{S}$, the equation:

$$(y_3 \circ y_{4,i_1}(x_{i_1})) \bullet (y_3 \circ y_{4,i_2}(x_{i_2})) \bullet \cdots \bullet (y_3 \circ y_{4,i_n}(x_{i_n}))$$

$$= y_3(y_{4,i_1}(x_{i_1}) \bullet y_{4,i_2}(x_{i_2}) \bullet \cdots \bullet y_{4,i_n}(x_{i_n})),$$

is satisfied. Therefore, one gets:

$$(y_{1,i_1} \circ y_2(x_{i_1})) \bullet (y_{1,i_2} \circ y_2(x_{i_2})) \bullet \cdots \bullet (y_{1,i_n} \circ y_2(x_{i_n}))$$

$$= y_3(y_{4,i_1}(x_{i_1}) \bullet y_{4,i_2}(x_{i_2}) \bullet \cdots \bullet y_{4,i_n}(x_{i_n})).$$

□

### 4.2. SAPKA-MK Version of SAA-5

SAA-5 can be described in the form of SAPKA-MK as:
The ingredients for this algorithm are:

$$I := \{i_1, i_2, \ldots, i_n\} \subset \mathbb{N},$$

$$\mathcal{P} := M(d, \mathbb{Z}_p),$$

and an operation $\bullet$ denotes the Schur-product, which is the element-wise matrix multiplication.

Here, $(\mathcal{P}, \bullet)$ forms a semigroup because for any $x, y \in \mathcal{P}$:

$$x \bullet y \in \mathcal{P},$$

and for any $x, y, z \in \mathcal{P}$:

$$x \bullet (y \bullet z) = (x \bullet y) \bullet z.$$

Denoting $L$ (resp. $R$) the left (resp. right) action of $M(d, \mathbb{Z}_p)$ on itself, defined by:

$$L_x(m) := (\prod_{b \in \{1, \cdots, d\}} x_{a,b}{}^{m_{b,g}}) \quad ; \quad R_x(m) := (\prod_{b \in \{1, \cdots, d\}} x_{b,g}{}^{m_{a,b}}) \quad ; \quad x, m \in M(d, \mathbb{Z}_p).$$

Then, additional ingredients are:

$$M_{\mathcal{P}} := \{L_x \ : \ x \in M(d, \mathbb{Z}_p)\} \cup \{R_x \ : \ x \in M(d, \mathbb{Z}_p)\},$$

$$\widehat{M}_{\mathcal{P}} := \{\text{invertible elements of } M_{\mathcal{P}}\}.$$

If Bob defines the six-tuple $(\{y_{1,j} \ ; \ j \in I\}, y_2, y_3, \{y_{4,j} \ ; \ j \in I\}, N_1, N_2)$ for each $j \in I$ as:

$$y_{1,j}(x) := \left( \prod_{b \in \{1, \cdots, d\}} (c^{\circ B_j x_B})_{b,g}^{(x)_{a,b}} \right) = R_{c^{\circ B_j x_B}}(x)$$

$$= c^{\circ x B_j x_B} \, ,$$

$$y_2 := id_{\mathcal{P}} \, ,$$

$$y_3(x) := \left( \prod_{b \in \{1,\cdots,d\}} x_{a,b}^{(x_B)_{b,g}} \right) = L_x(x_B) \, ,$$

$$y_{4,j}(x) := \left( \prod_{b\{1,\cdots,d\}} (c^{\circ B_j})_{b,g}^{(x)_{a,b}} \right) = R_{c^{\circ B_j}}(x) = c^{\circ x B_j} \, ,$$

$$N_1 := id_{\mathcal{P}} \, ,$$

$$N_2(x) := \left( \prod_{b \in \{1,\cdots,d\}} x_{a,b}^{(N_B^{-1})_{b,g}} \right) := L_x(N_B^{-1}) \, ,$$

and constructs the quadruple $(\{y'_{1,j} \; ; \; j \in I\}, y'_2, y'_3, \{y'_{4,j} \; ; \; j \in I\})$ as:

$$y'_{1,j} := y_{1,j} \circ N_1^{-1} \quad ; \quad y'_2 := N_1 \circ y_2 \quad ; \quad y'_3 := y_3 \circ N_2 \quad ; \quad y'_{4,j} := N_2^{-1} \circ y_{4,j} \, ,$$

for all $j \in I$, then Lemma 1 is achieved, i.e.,

$$y'_{1,j} \circ y'_2(x) = y_{1,j} \circ y_2(x) = c^{\circ x B_j x_B} =$$

$$c^{\circ x B_j N_B N_B^{-1} x_B} = y_3 \circ N_2 \circ N_2^{-1} \circ y_{4,j}(x) = y'_3 \circ y'_{4,j}(x) \, ,$$

and for $x, z \in \mathcal{P}$:

$$y'_3(x \bullet z) = \left( \prod_{b \in \{1,\cdots,d\}} (x \bullet z)_{a,b}^{(N_B^{-1} x_B)_{b,g}} \right) = \left( \prod_{b \in \{1,\cdots,d\}} x_{a,b}^{(N_B^{-1} x_B)_{b,g}} z_{a,b}^{(N_B^{-1} x_B)_{b,g}} \right)$$

$$= \left( \prod_{b \in \{1,\cdots,d\}} x_{a,b}^{(N_B^{-1} x_B)_{b,g}} \right) \bullet \left( \prod_{b \in \{1,\cdots,d\}} z_{a,b}^{(N_B^{-1} x_B)_{b,g}} \right) = y'_3(x) \bullet y'_3(z) \, .$$

By the formation of maps, each process of SAPKA-MK key agreement can be described.

### 4.3. SAA-5 without Schur-Exponentiations

In this section, we introduce a new PKA, which we call SAA-5 without Schur-Exponentiations (SAA-5-no-SE). SAA-5-no-SE can be described as follows.

The public parameters for this algorithm are:

$$d \in \mathbb{N} \quad ; \quad \text{a finite set } I \subset \mathbb{N} \, ,$$

$$\mathcal{P} := M(d, \mathbb{Z}_p) := \{d \times d\text{-matrices with entries in } \mathbb{Z}_p\} \, .$$

Denoting $L$ (resp. $R$) the left (resp. right) action of $M(d, \mathbb{Z}_p)$ on itself, defined by:

$$L_x(m) := xm \quad ; \quad R_x(m) := mx \quad ; \quad x, m \in M(d, \mathbb{Z}_p) \, .$$

Additional public parameters are:

$$M_{\mathcal{P}} := \{L_x \; : \; x \in M(d, \mathbb{Z}_p)\} \cup \{R_x \; : \; x \in M(d, \mathbb{Z}_p)\} \, ,$$

$$\widehat{M}_{\mathcal{P}} := \{\text{invertible elements of } M_{\mathcal{P}}\} .$$

Bob's secret ingredients are:

– $N_B, x_B \in M(d, \mathbb{Z}_p)$

– a set

$$\mathbb{B} := \{B_j \in \mathcal{P} \ : \ j \in I \ , \ \textbf{non-invertible}\}$$

**Step 1.** Bob defines, for each $j \in I$, the following six maps:

$$y_{1,j}(x) = xB_j x_B = R_{B_j x_B}(x) ,$$

$$y_2 = id_{\mathcal{P}} ,$$

$$y_3(x) = xx_B = R_{x_B}(x) ,$$

$$y_{4,j}(x) = xB_j = R_{B_j}(x) ,$$

$$N_1 = id_{\mathcal{P}} ,$$

$$N_2(x) = xN_B^{-1} = R_{N_B^{-1}}(x) .$$

From the six-tuple $(\{y_{1,j} \ ; \ j \in I\}, y_2, y_3, \{y_{4,j} \ ; \ j \in I\}, N_1, N_2)$, Bob constructs the quadruple $(\{y'_{1,j} \ ; \ j \in I\}, y'_2, y'_3, \{y'_{4,j} \ ; \ j \in I\})$ as:

$$y'_{1,j} := y_{1,j} \circ N_1^{-1} \quad ; \quad y'_2 := N_1 \circ y_2 \quad ; \quad y'_3 := y_3 \circ N_2 \quad ; \quad y'_{4,j} := N_2^{-1} \circ y_{4,j} ,$$

for all $j \in I$. With this construction, (12) is satisfied, i.e.,

$$y'_{1,j} \circ y'_2(x) = y_{1,j} \circ y_2(x) = xB_j x_B = xB_j N_B N_B^{-1} x_B = y_3 \circ N_1 \circ N_1^{-1} \circ y_{4,j}(x) = y'_3 \circ y'_{4,j}(x) .$$

Moreover, for any $x, z \in \mathcal{P}$, $y'_3$ satisfies:

$$y'_3(x + z) = y_3 \circ N_2(x + z) = (x + z)x_B N_B^{-1} = xx_B N_B^{-1} + zx_B N_B^{-1}$$

$$= y_3 \circ N_2(x) + y_3 \circ N_2(z) = y'_3(x) + y'_3(z) .$$

Thus, the multiple compatibility condition (9) is satisfied from Lemma 1.

Hence, for any choice of $x_j \in \mathcal{P}$ $(j \in I)$, following equation:

$$\sum_{j \in I} y'_{1,j} \circ y'_2(x_j) = y'_3 \circ \sum_{j \in I} y'_{4,j}(x_j) ,$$

is satisfied. Then, Bob prepares his secret key as a function:

$$x'_B := y_3 \circ N_2 = y'_3 , \tag{13}$$

and produces the public keys as following functions:

$$y_{B,1}(x_{i_1}, x_{i_2}, \dots, x_{i_n}) = \sum_{j \in I} y'_{1,j} \circ y'_2(x_j) = \sum_{j \in I} y_1 \circ N_1^{-1} \circ N_1 \circ y_2(x_j) = \sum_{j \in I} x_j B_j x_B , \tag{14}$$

$$y_{B,2}(x_{i_1}, x_{i_2}, \dots, x_{i_n}) = \sum_{j \in I} y'_{4,j}(x_j) = \sum_{j \in I} N_2^{-1} \circ y_{4,j}(x_j) = \sum_{j \in I} x_j B_j N_B . \tag{15}$$

**Step 2.** In order to send the public keys $(y_{B,1}, y_{B,2})$ to Alice, Bob sends the public matrices:

$$B_j x_B \quad , \quad B_j N_B \quad (j \in I) .$$

**Step 3.** Alice chooses as her secret key a set of matrices:

$$\{x_{A,j} \in M(d, \mathbb{Z}_p) \; ; \; j \in I\} \,,$$

and constructs her public key:

$$y_A = y_{B,2}(x_{A,i_1}, x_{A,i_2}, \ldots, x_{A,i_n}) = \sum_{j \in I} x_{A,j} B_j N_B \,.$$

**Step 4.** Alice sends the public key $y_A$ to Bob.

**Step 5.** The secret shared key (SSK) $\kappa$ is:

$$\kappa := \sum_{j \in I} x_{A,j} B_j x_B \,.$$

Alice knows the $B_j x_B$, so she can compute:

$$\kappa = y_{B,1}(x_{A,i_1}, x_{A,i_2}, \ldots, x_{A,i_n}) = \sum_{j \in I} x_{A,j} B_j x_B \,.$$

**Step 6.** Bob computes $\kappa$ as:

$$\kappa = x_3'(y_A) = x_3 \circ N_2 \Big( \sum_{j \in I} x_{A,j} B_j N_B \Big) = \Big( \sum_{j \in I} x_{A,j} B_j N_B \Big) N_B^{-1} x_B = \sum_{j \in I} x_{A,j} B_j x_B \,.$$

*4.4. The Comparison of Breaking Complexity between SAA-5 And SAA-5-no-SE*

The breaking complexity of SAA-5 is already evaluated. In this section, we assume 0 cost for solving the discrete logarithm problem, the same as [17] and Section 2.1. Under this assumption, we can prove the breaking complexity of SAA-5-no-SE is equivalent to that of SAA-5 by considering Eve's strategy to find the SSK of both algorithms.

**Theorem 1.** *The breaking complexity of SAA-5 is equivalent to that of SAA-5-no-E.*

**Proof.** (Strategy of Eve against SAA-5-no-SE)

Eve knows the following finite set of integer $I = \{i_1, i_2, \ldots, i_n\}$ where $|I| = n$ and matrices $B_j x_B$, $B_j N_B$, $y_A = \sum_{j \in I} x_{A,j} B_j N_B$ for all $j \in I$. She also knows following equation:

$$\kappa = \sum_{j \in I} x_{A,j} B_j x_B \,, \tag{16}$$

is held. She tries to recover $\kappa$ from public keys $B_j x_B$, $B_j N_B$, $y_A = \sum_{j \in I} x_{A,j} B_j N_B$ where all $B_j$, $x_B$, $N_B$ and $x_{A,j}$ are unknown for Eve.

(Strategy of Eve against SAA-5)

In this case, Eve's strategy to break the algorithm is that she gets the following logarithm of public keys as:

$$\log c^{\circ B_j x_B} = B_j x_B \log c \,,$$

$$\log c^{\circ B_j N_B} = B_j N_B \log c \,,$$

$$\log y_A = \sum_{j \in I} x_{A,j} B_j N_B \log c = \sum_{j \in I} x_{A,j} \log c^{\circ B_j N_B} \,,$$

for all $j \in I$. Then she try to recover $\log \kappa$, which is:

$$\log \kappa = \sum_{j \in I} x_{A,j} B_j x_B \log c = \sum_{j \in I} x_{A,j} \log c^{\circ B_j x_B} \, .$$

But, by putting $B'_j = B_j \log c$, Eve knows the following equations:

$$\log c^{\circ B_j x_B} = B'_j x_B \, , \tag{17}$$

$$\log c^{\circ B_j N_B} = B'_j N_B \, , \tag{18}$$

$$\log y_A = \sum_{j \in I} x_{A,j} \log c^{\circ B_j N_B} = \sum_{j \in I} x_{A,j} B'_j N_B \, , \tag{19}$$

and tries to recover

$$\log \kappa = \sum_{j \in I} x_{A,j} \log c^{\circ B_j x_B} = \sum_{j \in I} x_{A,j} B'_j x_B \, , \tag{20}$$

are held. She tries to recover $\log \kappa$ from public keys $B'_j x_B$, $B'_j N_B$, $y_A = \sum_{j \in I} x_{A,j} B'_j N_B$ where all $B'_j$, $x_B$, $N_B$ and $x_{A,j}$ are unknown for Eve. For Eve, this is the same strategy as against the SAA-5-no-SE case. □

### 4.5. Discussion: Performance of SAA-5-no-SE

Here, we estimate the computational complexity and report the performance test of SAA-5-no-SE. The total estimated number of multiplications for SAA-5-no-SE is given in Table 4. Notice that one matrix multiplication requires $d^3$ number of scalar multiplications. Although the calculation time is expected to be on the order of $d^3$, as in the SAA-5 case, it does not need extra $d^2 \log p$ number of multiplications, which are needed for the calculation of scalar exponentiations in SAA-5. We already know that the calculations of scalar exponentiations heavily effect the performance of SAA-5, so the calculation speed of it is expected to be much faster than SAA-5 with same condition.

**Table 4.** Key size and estimation of the time for multiplication of keys.

| Key | Bit Size | Steps |
|---|---|---|
| $y_A$ | $d^2 \log p$ | $(|I| - 1)d^3$ |
| $\kappa_A$ | $d^2 \log p$ | $(|I| - 1)d^3$ |
| $y_{B2}$ | $d^2 |I| \log p$ | $|I|d^3$ |
| $y_{B3}$ | $d^2 |I| \log p$ | $|I|d^3$ |
| $\kappa_B$ | $d^2 \log p$ | $2d^3$ |
| Total | | $4|I|d^3$ |

Here, we compare the calculation speed with SAA-5 for two settings. In the first setting, we measure the calculation speed of both algorithms for each SSK length from 800 bit to 16,000 bit, while $d$ and $|I|$ are both fixed as 10 to see the impact of $p$ and the SSK length on speed (results are in Table 5 and Figure 3). In the second setting, the calculation speed of SAA-5-no-SE for each SSK length from 512 bit to 5120 bit while $d = 8$ and $|I| = 5$ is measured, and has the same settings as in Table 2 and Figure 2. Since the SSK length of SAA-5-no-SE is given by $d^2 \log p$, which is the same number as SAA-5, the relation of size of prime numbers for D-H, SAA-5 and SAA-5-no-SE are described as:

$$\bar{\kappa} = p_{D-H} = 64 p_{SAA-5} = 64 p_{SAA-5-no-SE} \tag{21}$$

where $p_{SAA-5-no-SE}$ is the bit size of prime used in SAA-5-no-SE and $\bar{\kappa}$ is that of fixed SSK. Also, the computational complexity of SAA-5-no-SE (denoted by $CC_{SAA-5-no-SE}$) when the SSK length is fixed as $\bar{\kappa}$ is shown in Table 6 along with that of $SAA-5$ and D-H.

**Table 5.** Calculation speed for each SSK length while $d = 10$ and $|I| = 10$.

| SSK Length (bit) | $p$ (bit) | SAA-5 | SAA-5-no-SE |
|---|---|---|---|
| 800 | 8 | 34.42 | 16.88 |
| 1600 | 16 | 40.26 | 9.58 |
| 2400 | 24 | 44.98 | 9.92 |
| 3200 | 32 | 52.36 | 12.86 |
| 4000 | 40 | 89.82 | 19.24 |
| 4800 | 48 | 75.18 | 18.78 |
| 5600 | 56 | 83.94 | 14.74 |
| 6400 | 64 | 101.16 | 14.76 |
| 7200 | 72 | 145.04 | 15.62 |
| 8000 | 80 | 150.58 | 14.68 |
| 8800 | 88 | 164.84 | 17.64 |
| 9600 | 96 | 176.44 | 17.68 |
| 10,400 | 104 | 191.20 | 17.46 |
| 11,200 | 112 | 202.76 | 17.22 |
| 12,000 | 120 | 213.36 | 15.96 |
| 12,800 | 128 | 223.30 | 16.54 |
| 13,600 | 136 | 314.08 | 18.24 |
| 14,400 | 144 | 336.24 | 17.58 |
| 15,200 | 152 | 349.96 | 18.02 |
| 16,000 | 160 | 364.66 | 20.30 |

**Table 6.** Computational complexity of each algorithm when SSK length is $\bar{\kappa}$.

| $CC_{D-H}$ | $CC_{SAA-5}$ | $CC_{SAA-5-no-SE}$ |
|---|---|---|
| $4\bar{\kappa}$ | $4|I|(d^3 + \bar{\kappa})$ | $4|I|d^3$ |

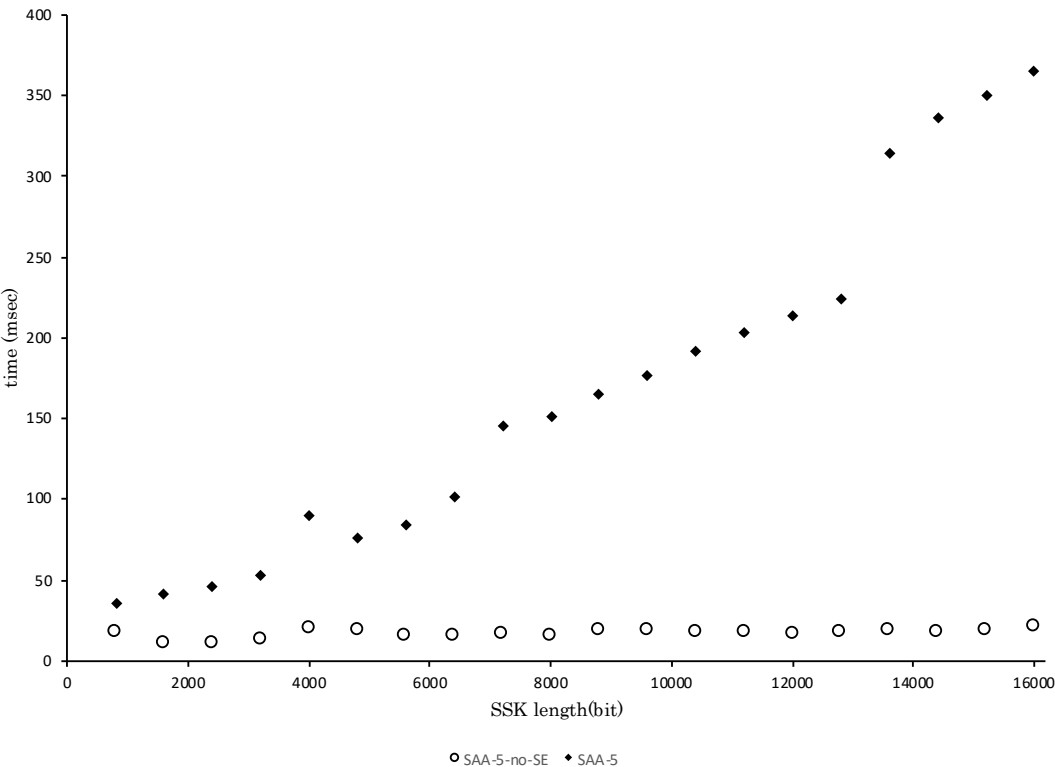

**Figure 3.** Calculation speed for each SSK Length with strongly-asymmetric algorithm (SAA-5) while $d = 10$ and $|I| = 10$.

### 4.6. Experimental Result: Performance of SAA-5-no-SE

In Figure 3, the white circle indicates the time spent to calculate in SAA-5-no-SE, and the black diamond indicates the time in SAA-5. As we expected, we can see the prime $p$ does not effect the calculation speed of SAA-5-no-SE. Moreover, the calculation time of SAA-5-no-SE is not heavily influenced by its SSK length.

In Table 7 and Figure 4, the white circle indicates the time spent to calculate in SAA-5, and the black circle indicates the time in SAA-5-no-SE. Note that the calculation time of SAA-5 shown in Figure 4 is the value of Table 2 and Figure 2. With Table 2 and Figure 2, we can check SAA-5-no-SE out-performs not only SAA-5 of all SSK lengths but also D-H of over 1536 bit. Especially when the SSK lengths are 5120 bit, SAA-5-no-SE is approximately 100 times faster than D-H.

**Table 7.** The time to generate SSK.

| SSK Length (bit) | SAA-5-no-SE(msec) |
| --- | --- |
| 512 | 8.32 |
| 1024 | 5.63 |
| 1536 | 4.57 |
| 2048 | 5.26 |
| 2560 | 7.41 |
| 3072 | 6.84 |
| 3584 | 6.15 |
| 4096 | 4.80 |
| 4608 | 5.75 |
| 5120 | 6.02 |

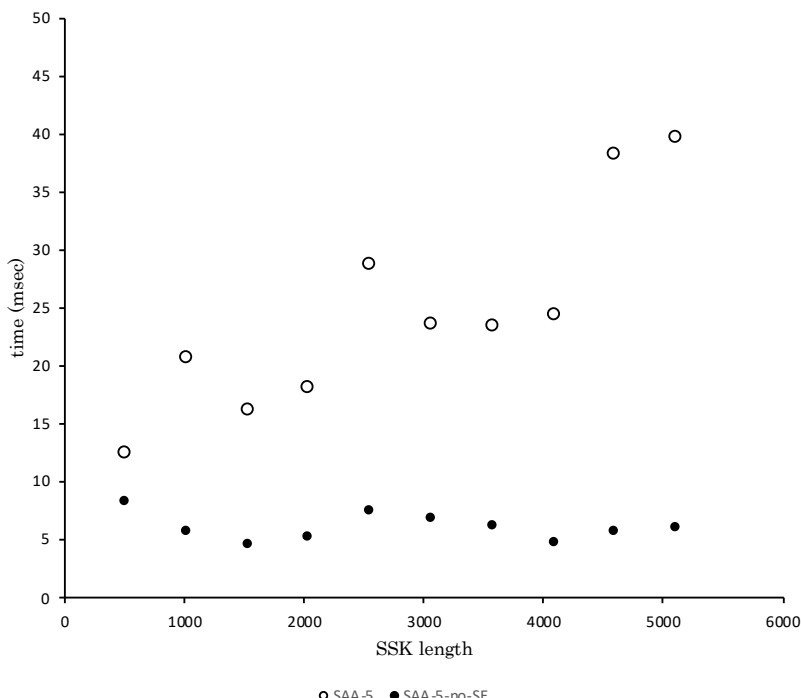

**Figure 4.** Comparison of the generation time of the SSK with SAA-5.

As for performances of other standard PKA/asymmetric cryptography, such as ElGamal, RSA and Elliptic curve D-H, there are not many differences with that of D-H (see Table 1 of [20], Figures 1–3 of [21] and Figures 3, 4 of [18]) and the speed of their algorithms increases exponentially as the SSK length increases. Although some algorithms might be faster than SAA-5 or SAA-5-no-SE, especially for short key lengths, the calculation speed of SAA-5 and SAA-5-no-SE increase linearly, not exponentially. This is one of the advantage points of SAA-5 and SAA-5-no-SE.

## 5. Conclusions

In this paper, we showed that the performance of SAA-5 is much more effective than D-H, especially for large keys, because of its algebraic property and variety of parameters.

Moreover, a more general scheme of SAA-5, called SAPKA-MK, is proposed. SAA-5 is shown to be described in the form of SAPKA-MK, and a new PKA algorithm called SAA-5-no-SE is introduced as a concrete example of SAPKA-MK. Alice's secret key structure of SAA-5-no-SE is the same as that of

SAA-5, but the functions used in the public key and SSK generation steps are different in several ways. Our performance test on SAA-5-no-SE showed its calculation time increases linearly not exponentially as the SSK length increases. The test also showed that SAA-5-no-SE is a much more efficient PKA algorithm than not only D-H of over 1536bit but also the SAA-5 of all SSK lengths, and its security level is as high as SAA-5.

**Author Contributions:** Conceptualization, K.J., S.I. and M.R.; formal analysis, K.J. and S.I.; software, M.R.; writing—original draft, K.J. and S.I.; writing—review and editing, K.J., S.I. and M.R. All authors have read and agreed to the published version of the manuscript.

**Funding:** This research received no external funding.

**Conflicts of Interest:** The authors declare no conflict of interest.

## Abbreviations

The following abbreviations are used in this manuscript:

| | |
|---|---|
| PKA | public key agreement |
| D-H | Diffie-Hellman |
| SAPKA | strongly-asymmetric public key agreement |
| SAA | strongly-asymmetric algorithm |
| SSK | secret shared key |

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
