# Peer review of "Implementation of a New Strongly-Asymmetric Algorithm and Its Optimization"

_cryptography, doi:10.3390/cryptography4030021_

Round 1

Reviewer 1 Report

The authors extend their previous work by adding experimental results and comparing with DH key agreement.  The authors should make their experiments and comparison more convincing by adding at least the following three aspects:

  1. clearly indicate the complexity of DH key agreement protocol. 
  2. clearly indicate the relationship between the parameters of SAA-5 and those of DH key agreement.
  3. Since the total complexity is  4|I|(d^3 + d^2 log(p)), and the authors reasoned and validated its efficiency, which is good. However, on the other hand, such a low complexity may result in low complexity of brute-force attacks correspondingly.   Authors need to prove or at least justify the reverse.   

Author Response

Dear reviewer 1

We express appreciation to you giving us the fruitful comments.
According to your comments, we revised the article as following:

1. clearly indicate the complexity of DH key agreement protocol. 

>> In the section 3.2, we added the computational complexity of D-H denoted by CC_{D-H}.
When the bit length of SSK is described as ¥hat k, then the bit size of each secret key and public key for Alice (Bob) is also ¥hat k. the scalar exponentiation g^{x_A} require log x_A steps of scalar multiplication, which is nearly equal to the bit size of x_A. Thus, nearly 4 ¥hat k steps of scalar multiplication are needed for D-H.

2. clearly indicate the relationship between the parameters of SAA-5 and those of DH key agreement.

>> We added explanation about the relationship of parameters of SAA-5 and those of D-H in section 3.3. In the experiment, the bit size of prime number (used for modulo operation) of each algorithm is depend on the fixed SSK length. For example, when SSK length is fixed as 1024 bit, the bit size of prime for SAA-5 is 16bit and that for D-H is same bit size as SSK. Also we added table (table 2) which indicates bit size of prime of both algorithms for each SSK length.

3. Since the total complexity is  4|I|(d^3 + d^2 log(p)), and the authors reasoned and validated its efficiency, which is good. However, on the other hand, such a low complexity may result in low complexity of brute-force attacks correspondingly.   Authors need to prove or at least justify the reverse.   

>> In section 2.1, we tried the brute-force attack against SAA-5. Eavesdropper has to search one particular matrix pair (x_1 log c, x_4) within the extremely large set is shown (see (8)). Please refer to [14] in which security analysis is also done. In section 4.4, the breaking complexity of SAA-5-no-SE is shown to equivalent to that of SAA-5 under when 0 cost for solving discrete logarithm problem is assumed.

Best regards,
Koki Jimbo
Satoshi Iriyama
Massimo Regoli

2020, 22 July

Reviewer 2 Report

This article is about the implementation of new asymmetric
algorithms. The authors analyze, among others SAA-5 algorithm and compare it with the DH protocol. Besides, the authors discuss their modifications and optimization.

The topic of the article is interesting and is part of current research on broadly understood methods of securing communication. The text is easy to read from the language side, although some issues require a comment from the authors:
1) According to the guidelines of the journal, the abstract is too long.
2) The Introduction part is quite modest in information about asymmetric cryptography and can undoubtedly be more extended with other public-key algorithms.
3) The article in the title has the sentence "implementation of the algorithms". Thus, it can be understood as the actual implementation of the presented algorithm in one of the common programming languages, e.g., Java. However, the authors only present the implementation of the DH key exchange protocol. Maybe it would be worth using an Appendix or an external medium such as GitHub to show the algorithms' implementation.
4) Java code implementing the DH protocol should also move to the Appendix.
5) It may be worth improving the terminology of mathematical symbols, e.g., function names using symbols of type y instead of x. It is also worth defining each of the customs operations precisely, e.g., empty circle in formula (8).
6) After the formulas, I suggest using punctuation marks consistently.
7) The authors compare the presented algorithm only with the DH protocol. It would be worth seeing how it looks on the background of other commonly used cryptographic methods.
8) The References list is far too short.

Author Response

Dear reviewer 2

We express appreciation to you giving us the fruitful comments.
According to your comments, we revised the article as following:

1) According to the guidelines of the journal, the abstract is too long.

>> We reduced abstract as around 200 words long.

2) The Introduction part is quite modest in information about asymmetric cryptography and can undoubtedly be more extended with other public-key algorithms.
8) The References list is far too short.

>> We extended the introduction including other public-key algorithms, the topic of potential threat of Shor’s quantum algorithm and the topic about post quantum algorithms such as lattice based one. References list is increased as a result of the extension.

3) The article in the title has the sentence "implementation of the algorithms". Thus, it can be understood as the actual implementation of the presented algorithm in one of the common programming languages, e.g., Java. However, the authors only present the implementation of the DH key exchange protocol. Maybe it would be worth using an Appendix or an external medium such as GitHub to show the algorithms' implementation.
4) Java code implementing the DH protocol should also move to the Appendix.

>> We decided to upload the Java codes of D-H, SAA-5 and SAA-5-no-SE into Github instead of making Appendix because presenting all source codes will be too long for the manuscript. So, we deleted the description of Java code implementing of D-H.
Please check: https://github.com/jimbobmij/project_KSM

5) It may be worth improving the terminology of mathematical symbols, e.g., function names using symbols of type y instead of x. It is also worth defining each of the customs operations precisely, e.g., empty circle in formula (8).

>>We changed function names from x to y. And we added precise denotations for every symbols including empty circle of formula (9) you mentioned.

6) After the formulas, I suggest using punctuation marks consistently.

>>We added added commas or periods to all equations.

7) The authors compare the presented algorithm only with the DH protocol. It would be worth seeing how it looks on the background of other commonly used cryptographic methods.

>>We have not tested the performance of other algorithms. But we added a few references in which the performance of standard algorithms such as RSA, ElGamal, EC-DH are tested ([16],[17]). We can see the calculation time of those algorithms increase exponentially as SSK length increased from the references. Because the point we want to claim most is that the time of SAA-5 and SAA-5-no-SE increase linearly, not exponentially (can be checked from figure 4 and 5) so, we thought tests for not all of standard algorithms were necessary.

Best regards,
Koki Jimbo
Satoshi Iriyama
Massimo Regoli

2020, 22 July

Round 2

Reviewer 1 Report

The relationship between the parameters  and the complexity of DH and the new method can be more detailed. In addition, discussion about the man-in-the-middle attack of the new method should be done. 

Author Response

Dear reviewer 1

We express appreciation to you giving us the fruitful comments.
According to your comments, the point we modified are:

The relationship between the parameters  and the complexity of DH and the new method can be more detailed.

>> the relationship of parameters and computational complexity between D-H and the proposed algorithm(SAA-5-no-SE) are added in section 4.5 of relation (21) and table 5. When the length of SSK are fixed as ¥bar{¥kappa} for both D-H and SAA-5-no-SE, the size of prime number is the only related parameter between them. In this case, the size of prime for D-H (denoted by p_{D-H}) is:
   p_{D-H} = ¥bar{¥kappa}
and that for SAA-5-no-SE (denoted by p_{SAA-5-no-SE}) is:
   p_{SAA-5-no-SE} = ¥bar{¥kappa}/d^2
Moreover, the computational complexity of SAA-5-no-SE is given by 4|I|d^3, namely, not depends on the size of prime (this can be checked from figure 3, 4). This is not the case for SAA-5 and D-H (please refer table 5 and figure 2,3,4).

And one answer for the comment:

In addition, discussion about the man-in-the-middle attack of the new method should be done. 

>> In some sense, the proposed algorithm and SAA-5 provide just faster and more secure (excluding against man-in-the-middle attack) key agreement than D-H does. Thus, to be used in practice, schemes such as digital signature with trusted third parties will be needed for them as well as when D-H is used. We are now trying to extend the SAPKA-MK(introduced in section 4.1) framework to include digital signature and asymmetric cryptography scheme to find a solution for man-in-the-middle-attack threat.  But it is not yet finished and it will be new algorithms which need another security analysis, so discussion for man-in-the-middle-attack will be in our sequel paper, not here.

Best regards,
Koki Jimbo
Satoshi Iriyama
Massimo Regoli

2020, 28 July

Reviewer 2 Report

It's my second review of an article by Jimbo et al. The authors responded to the previous review - the answers are considered sufficient. Thanks to the introduced changes, the article's quality, both in the introductory part and the research part, is much better. One - in my opinion - a drawback that has not been properly corrected is the too-short set of items in References. In this regard, it is possible to further improve the article into items on, for example, other current proposals for public-key algorithms.

Author Response

Dear reviewer 2

We express appreciation to you giving us the fruitful comment.
According to your comment, the point we modified is:

a drawback that has not been properly corrected is the too-short set of items in References. In this regard, it is possible to further improve the article into items on, for example, other current proposals for public-key algorithms.

>> We added three references [11], [13], [14]. In [11], typical 26 current public key algorithms are listed. Because algorithms such as [12], [13] and [14] utilise matrix for maintain their exhaustive robustness, we added them in the references expecting that there might be some relations with our models. Clarifying the relations and comparing efficiency and security with them are not yet discussed but, will be one of our future main themes.

Best regards,
Koki Jimbo
Satoshi Iriyama
Massimo Regoli

2020, 28 July